# Do Aging and Parity Affect VEGF-A/VEGFR Content and Signaling in the Ovary?—A Mouse Model Study

**DOI:** 10.3390/ijms24043318

**Published:** 2023-02-07

**Authors:** Valentina Di Nisio, Gianna Rossi, Alessandro Chiominto, Ezio Pompili, Sandra Cecconi

**Affiliations:** 1Department of Clinical Science, Intervention and Technology, Division of Obstetrics and Gynecology, Karolinska Institutet and Karolinska University Hospital, 14186 Huddinge, Sweden; 2Department of Life, Health and Environmental Sciences, University of L’Aquila, Via Vetoio, 67100 L’Aquila, Italy; 3Department of Pathology, San Salvatore Hospital, 67100 L’Aquila, Italy

**Keywords:** VEGF-A, VEGFR2, aging, ovary, parity

## Abstract

In this study, the effects of aging and parity on VEGF-A/VEGFR protein content and signaling in the mice ovaries were determined. The research group consisted of nulliparous (virgins, V) and multiparous (M) mice during late-reproductive (L, 9–12 months) and post-reproductive (P, 15–18 months) stages. Whilst ovarian VEGFR1 and VEGFR2 remained unchanged in all the experimental groups (LM, LV, PM, PV), protein content of VEGF-A and phosphorylated VEGFR2 significantly decreased only in PM ovaries. VEGF-A/VEGFR2-dependent activation of ERK1/2, p38, as well as protein content of cyclin D1, cyclin E1, and Cdc25A were then assessed. In ovaries of LV and LM, all of these downstream effectors were maintained at a comparable low/undetectable level. Conversely, the decrease recorded in PM ovaries did not occur in the PV group, in which the significant increase of kinases and cyclins, as well phosphorylation levels mirrored the trend of the pro-angiogenic markers. Altogether, the present results demonstrated that, in mice, ovarian VEGF-A/VEGFR2 protein content and downstream signaling can be modulated in an age- and parity-dependent manner. Moreover, the lowest levels of pro-angiogenic and cell cycle progression markers detected in PM mouse ovaries sustains the hypothesis that parity could exert a protective role by downregulating the protein content of key mediators of pathological angiogenesis.

## 1. Introduction

Many women are delaying pregnancy into their 30s and beyond due to either social or personal reasons [1,2], or because of infertility [3,4]. The lack of pregnancy could have many repercussions on their health, including increased risk of gynecological cancers, particularly endometrial [5], breast [6], and ovarian [7,8] cancers. Indeed, pregnancy and childbearing at a younger age has been proven to exert a protective role against breast cancer in the long term [9,10], except the predisposition linked to *BRCA1/BRCA2* mutations [11] or to family history [12]. Conversely, aging [13], older menopausal age [14], and nulliparity [15] are considered as predisposing factors. In particular, nulliparity has been correlated with high Ki67 and cyclin D1 levels in HER2-positive breast tumors [16]. Epidemiological data confirm that parity can exert a protective role also against ovarian cancer (OC) [17], which usually increases with age, predominantly in postmenopausal women [18,19], and favors long-term survival after diagnosis [20,21].

In our previous study [22], we found that in young fertile female mice, VEGF-A and VEGFR2 protein content can be differentially modulated by parity and nulliparity. In fact, nulliparity could stimulate the formation of an ovarian microenvironment favoring proangiogenic alterations. Although angiogenesis is an extremely complex mechanism that involves a great number of factors and molecular pathways, it has been plentifully demonstrated in both physiological and pathological settings that VEGF-A is the angiogenic key factor that stimulates endothelial cell proliferation, promotes cell migration, and induces the stabilization of blood vessels, which is a fundamental process in vascular development [23]. Briefly, this growth factor, particularly isoforms 164 and 120 that are responsible for angiogenic mechanisms in the mouse ovary [22], binds predominantly two of its receptors, i.e., VEGFR1 and VEGFR2, triggering downstream ERK1/2 and AKT signaling. Therefore, research regarding angiogenesis related processes and putative disturbances focuses on the VEGF/VEGFR signaling modulation [24]. Here, we investigated whether pregnancy and nulliparity could differently impact VEGF-A/VEGFR2 expression also during aging by assessing the protein content of VEGF-A and of its receptors (VEGFR1 and 2) in ovaries of late-reproductive (L; 9–12 months old) and post-reproductive (P; 15–18 months old) multiparous (M) and nulliparous (V) mice. Moreover, we assessed the protein content of markers related to VEGFR2-dependent proliferation and migration signaling pathways (ERK1/2 and p38), and cell cycle regulators (cyclin D1, cyclin E1, and Cdc25A).

## 2. Results

### 2.1. VEGF and VEGFR Tissue Localization and Protein Content in Whole Ovaries from Late and Post-Reproductive Mice

Comparable ovarian VEGF-A 164 and 120 protein contents were present in the two groups of late reproductive (L) mice independently of parity/nulliparity (LM and LV: *p* > 0.05; Figure 1A,B). In turn, an overall decrease in VEGF-A content was recorded in the ovaries of older mice (LM, LV vs. PM, PV: *p* < 0.05). The most noticeable decline was observed in PM mice, since in their ovaries, VEGF-A 164 and VEGF-A 120 were undetectable or lowly-expressed, respectively (Figure 1A,B). On the contrary, in PV ovaries, VEGF-A 120 content was apparently unaffected by aging, while that of VEGF-A 164 was significantly reduced (nearly two-fold), compared to the younger counterparts (PV vs. LV: *p* < 0.05; Figure 1A,B).

VEGFR1 and VEGFR2 contents were similar in all experimental groups of mice (Figure 1A,C,D; *p* > 0.05), but VEGFR2 phosphorylation occurred less efficiently in PM mice, thereby mirroring the trend described for VEGF-A (PM vs. LM, PV: *p* < 0.05; Figure 1E).

Based on previous results, IHC analysis was performed to detect VEGF-A and p-VEGFR2 tissue localization in PM and PV mouse ovaries. In PM ovaries, VEGF-A was present in follicles, blood vessels, and stromal compartment (Figure 2); in PV ovaries, a stronger immunoreactivity for VEGF-A was detected also in cells of ovarian surface epithelium (OSE) and endothelial cells surrounding the lumen of blood vessels, in comparison with PM (Figure 2A,B). Moreover, in PV ovaries, an increased number of small blood vessels was detectable (Figure 2B, circles), together with the presence of multi-layered OSE cells, in comparison with the PM experimental group (Figure 2B, white arrowheads). Results from p-VEGFR2 immunolocalization mirrored that of VEGF-A in both PM and PV ovaries (Figure 2C,D).

### 2.2. VEGFR2 Signaling Pathway Activation

Total and phosphorylated ERK1/2 contents were expressed almost exclusively in V mice and significantly higher in the ovaries of the older group (PV vs. LV: ERK1/2 = +57%; p-ERK1/2 = +98%), in comparison with the other ovarian samples (PV vs. PM, LV, LM: *p* < 0.001; Figure 3A–C). It should be noted that p38 kinase (entirely phosphorylated) was reported only in PV mice, indicating either a lack of the target protein in the other groups or a too low detection capability of the used method (PV vs. PM, LV: *p* < 0.05; Figure 3A,D,E).

Since ERK1/2 can activate multiple targets involved in cell cycle regulation [25], protein levels of cyclin D1, cyclin E1, and Cdc25 phosphatase were determined. As shown in Figure 4, ERK1/2-dependent stimulation of cyclin D1, cyclin E1, and Cdc25A protein content was differentially modulated by parity and aging. Protein contents were similar in younger mice (LM vs. LV: *p* > 0.05, Figure 4). However, the protein content of cyclin D1 decreased significantly in both aged groups, and in particular in PM ovaries (PV vs. PM, PV vs. LV, PM vs. LM: *p* < 0.05; Figure 4B). Cyclin E1 and Cdc25A were highly expressed in PV ovaries, in comparison with other experimental groups (PV vs. LV, PV vs. PM: *p* < 0.05; Figure 4C,D). It is noteworthy that in PM ovaries, their contents decreased, and especially the protein content of cyclin E drastically declined (PM vs. PV: −95%; *p* < 0.001; Figure 4C).

## 3. Discussion

In this study, the effects of aging and parity on VEGF-A/VEGFR expression and signaling in the mice ovaries have been described. In fact, we found that the VEGF-A protein content, and especially of VEGF-A 164 isoform, and of pVEGFR2 were significantly lower in the group of post-reproductive multiparous mice (PM). Moreover, we described a parity- and age-dependent protein content decrease of ERK1/2 and p38 kinases and of cell cycle regulators cyclins D1 and E1, as well as of Cdc25A. Our data demonstrated that, during aging, parity can modulate the physiological expression of the proangiogenic factor VEGF-A and the activation of its receptor VEGFR2, by avoiding the dysregulation of this pathway. Our results are consistent with those of Hou and collaborators [26], who identified aging and multiparity as factors that can respectively impact and delay the tumorigenesis of high-grade serous carcinoma in genetically engineered BPRN mice.

Activation of VEGFR2 signaling has been extensively documented in the process of normal ovarian angiogenesis [27], while the abnormal regulation of this stimulus causes the onset of several pathological conditions [28,29]. The high expression of VEGF-A found in OSE cells might be an early sign of altered tissue morphology, finally leading to the malignant transformation of these cells [30]. As expected, in ovarian sections, both VEGF and p-VEGFR2 were localized in the endothelial cells surrounding the lumen of blood vessels that appeared more numerous in the ovaries of the PV experimental group. Moreover, both VEGF-A and p-VEGFR2 were found to be significantly downregulated in ovaries of PM mice, indicating that the strong reduction of the proangiogenic signaling is mainly related to the parity status. Since a significant downregulation of VEGF has been detected in the menstrual blood of multiparous women, in comparison with nulliparous women [31], it is likely that parity could repress, in a still unknown manner, the synthesis of this growth factor. Therefore, as already hypothesized in our previous work [22], the formation of a proangiogenic environment in the ovaries could create a favorable environment for the onset of vascular remodeling [32].

Since altered VEGFR2 activation in PV ovaries could predispose endothelial cells to proliferation and migration [27,33], we assessed the protein content of ERK1/2 and p38 kinases. Whilst p-VEGFR2 did not over-activate such a downstream signaling in ovaries of LV mice, in PV mice, both ERK1/2 and p38 contents were significantly increased. Notably, while total ERK1/2 protein content appears to be closely related to nulliparity due to its overexpression in both LV and even more in PV mice, p38 is overexpressed only in PV ovaries. Currently, it has been reported that ononin, a flavonoid detected in food and plants, suppressed angiogenesis via the downregulation of ERK1/2 and VEGFR2 in HUVEC cells [34]. These findings reinforce the key role of ERK1/2 as a pro-angiogenic driver in different animal models [35,36,37]. In line with our results, VEGFR2-dependent cell proliferation could be mainly addressed by p-ERK1/2 and also supported by the higher levels of p-p38, as documented for other mammalian cells [38,39].

To determine whether an abnormal kinase activation would stimulate the proliferative process, we investigated the protein content of cyclins D1 and E1. In fact, it has been recently reported that a proangiogenic environment can promote proliferation of endothelial cells via the ERK/cyclin D1 axis [40,41]. Moreover, literature data report an increase in cyclin E1 levels through VEGF-A dependent stimulation of ERK pathway [42,43]. The finding that cyclin D1 and E1 overexpression in the ovaries of PV mice occurs concomitantly with increased phosphorylation levels of ERK1/2, suggests that the VEGF-A/VEGFR2 signaling pathway could be improperly activated towards an increased cell proliferation. This hypothesis is strengthened also by the higher levels of Cdc25A, which normally promotes the G1/S transition [44], detected in the same ovaries. Notably, also in normal non-transformed cells, the overproduction of cyclin E, along with Cdc25A, is responsible for perturbing normal DNA replication and induction of genome instability [45,46]. In this context, it is worth mentioning that the overexpression of both cyclin D1 and E1 can be considered as a marker of oncogenesis for several tumor types, including OC [47,48]. Interestingly, in our samples, a great increase of cyclin E1 content in PV, compared to PM ovaries, was detectable, and even if the mechanisms leading to its deregulated expression are still unknown, we can consider this rise as a possible alarming marker of an altered cell cycle. Nevertheless, the levels of cyclin D1 in PV mice, even if doubled, in comparison with PM, are lower, compared to LV, the younger counterpart. This difference suggests an attempt from the ovarian environment to block the fast G1/S transition, as described in OC cells in which the increased protein content of cyclin D1 lead to the blockade of cell cycle transition despite the high levels of cyclin E [49].

In conclusion, our results show that pregnancy may exert a protective role on the ovarian environment by downregulating the VEGF-A/VEGFR2-dependent proangiogenic signaling and the protein content of some of the proteins controlling cell cycle progression. To the best of our knowledge, this is the first time that such interplay between angiogenic and cell proliferation pathways is described in ovaries from old multiparous and nulliparous mice. The finding that ovarian tissue started showing signs of OSE cell proliferation and increment in the microvasculature, reinforces the hypothesis that aging and nulliparity could promote a pro-angiogenic environment prone to uncontrolled cell proliferation, as demonstrated by the deregulation of the normal protein content of cyclins D1 and E1. In this context, our results are in line with epidemiologic data that correlate increased risk of OC with both age and nulliparity [17,50]. It could be of interest to better understand how aging- and nulliparity-related modifications of angiogenic signaling could contribute to determine the onset of OC, together with other genetic and environmental factors.

## 4. Materials and Methods

### 4.1. Chemicals

Chemicals used in this study were purchased from the following sources: rabbit polyclonal VEGF-A (sc-507), p38 (sc-7149), cyclin D1 (sc-753), cyclin E (sc-481) and actin (sc-1616R); mouse monoclonal Flt-1 (sc-271789; VEGFR1), Flk-1 (sc-6251; VEGFR2), phospho-ERK 1/2 (Thr202/Tyr204; sc-16982-R), ERK 1/2 (sc-135900), Cdc25A (sc-7389) and GAPDH (sc-32233); secondary antibody goat anti-mouse IgG conjugated to HRP (sc-2005) from Santa Cruz Biotechnology (Santa Cruz, CA, USA). Rabbit monoclonal phospho-VEGFR2 (Tyr1175; #2478) and rabbit polyclonal phospho-p38 (Thr180/Tyr182; #9211) were purchased from Cell Signaling Technology (Beverly, MA, USA). Secondary antibody goat anti-rabbit IgG conjugated to horseradish peroxidase (HRP), (cat. 111-035-003) was obtained from ThermoFisher Scientific (Waltham, MA, USA), and ECL Star-Enhanced chemiluminescent substrate from Cyanagen (Bologna, Italy). Mouse to mouse HRP ready-to-use kit was obtained from ScyTek Laboratories, Inc. (Logan UT, USA). All of the other reagents were purchased from Sigma-Aldrich (St. Louis, MO, USA), and were of the purest analytical grade.

### 4.2. Animals and Sample Collection

*Mus musculus* Swiss CD1 female mice (Harlan Italy, Udine, Italy) were housed in an animal facility under controlled temperature (21 ± 1 °C) and light (12 h light/day) conditions, with ad libitum access to food and water.

Mice of the same age (2 months-old, *n* = 80) were either mated (2 consecutive gestation cycles) with males of proven fertility (*n* = 40) or not, thus forming the group of multiparous (M; *n* = 40) and nulliparous virgin (V; *n* = 40) mice, respectively. Both M and V mice were then aged and further sorted into 2 groups: late-reproductive (L) mice (9–12 months-old) indicated as LM and LV, and post-reproductive (P) mice (15–18 months-old) indicated as PM and PV according to Asano [51]. When they reached the selected age, the mice were euthanatized, and whole ovaries were either stored at −196 °C under liquid nitrogen for further analysis or fixed in 4% paraformaldehyde overnight (o.n.) at 4 °C for paraffin embedding.

Experiments involving animals and their care were performed in conformity with national and international laws and policies (European Economic Community Council Directive 86/609, OJ 358, Dec 12, 1987; Italian Legislative Decree 116/92, Gazzetta Ufficiale della Repubblica Italiana n. 40, 18 February 1992; National Institutes of Health Guide for the Care and Use of Laboratory Animals, NIH publication no. 85-23, 1985). This project was approved by the internal ethics committee of the University of L’Aquila (2018). All efforts were made to minimize suffering. The method of euthanasia consisted of an inhalant overdose of carbon dioxide (CO_2_, 10–30%), followed by cervical dislocation.

### 4.3. Western Blotting

Whole ovaries were immersed in lysis buffer (50 mM Tris, pH 7.4, 150 mM NaCl, 1 mM EDTA, and 1% Igepal) containing protease inhibitors (1 mM phenylmethylsulphonylfluoride, 1 μg/mL leupeptin, and 1 μg/mL aprotinin) and phosphatase inhibitors (1 mM sodium fluoride, 10 mM sodium pyrophosphate, and 1 mM sodium orthovanadate), homogenized using a rotor stator tissue homogenizer (Precellys 24, Bertin Technologies; 2 cycles of 10 s at 5000× *g*) and centrifuged. Protein concentration was determined by Bio-Rad protein assay (Bio-Rad Laboratories, Inc., Hercules, CA, USA). Sixty μg of protein/whole ovaries was loaded onto 8% or 12% gels under reducing conditions, except for VEGFR1 and VEGFR2 that were examined in a nonreducing condition. Following transfer, blots were incubated with anti-VEGF-A (1:200), anti-VEGFR1 (1:1000), anti-VEGFR2 (1:200), anti-p-VEGFR2 (1:1000), anti-p-ERK1/2 (1:200), anti-ERK1/2 (1:200), anti- p-p38 (1:250), anti-p38 (1:250), anti-cyclin D1 (1:200), anti-cyclin E (1:200), anti-Cdc25A (1:200) and antibodies o.n. at 4 °C. Unfortunately, we could not assess VEGFR1 phosphorylation status due to the lack of commercially available antibodies targeting it.

Secondary antibodies conjugated to HRP, i.e., goat anti-rabbit (1:5000) and goat anti-mouse (1:5000), were incubated for 1 h at room temperature (r.t.). Then, peroxidase activity was detected using a ECL Star-Enhanced chemiluminescent substrate. The nitrocellulose membranes were examined using the Alliance LD2-77WL imaging system (Uvitec, Cambridge, UK). Densitometric quantification was performed with the public-domain software NIH Image V.1.62 and standardized using actin and/or GAPDH as loading controls. The signals of p-VEGFR2, p-ERK1/2, and p-p38 were normalized to the respective total of VEGFR2, ERK1/2, and p38, as previously described [22,52].

### 4.4. Hematoxylin-Eosin (H&E) and Immunohistochemistry (IHC)

Whole ovaries were processed and stained for H&E evaluation, as previously described [22]. Briefly, after fixation in 4% formalin, ovaries were embedded in paraffin, sectioned (4 µm/section), stained, and mounted. Sections were examined using StereoZoom^®^ Leica S8 APO and images were acquired with a Leica EC3 camera.

To determine the localization of VEGF-A and p-VEGFR2, IHC was performed, as briefly described below. Whole mouse ovaries were embedded in paraffin and sectioned (4 μm); after deparaffinization, the sections were re-hydrated, treated with 10 mM sodium citrate (pH 6.0), and washed three times in phosphate buffered saline (PBS) for 5 min. Mouse-to-mouse HRP ready-to-use Kit was used according to the manufacturer’s instructions. Sections were then incubated at 4 °C o.n. in humidified chamber with the following primary antibody: anti-VEGF-A (1:50), anti-p-VEGFR2 (1:100). Slides utilized as negative control were incubated with 3% BSA. Ovarian sections from all of the experimental groups were stained simultaneously for both markers to avoid any technical bias. Following washing with PBS, the slides were incubated with Ultra Tek anti-polivalent of the mouse-to-mouse kit for 15 min at r.t.. Following washing with PBS, the slides were incubated with Ultra Tek HRP, washed again with PBS, and incubated with DAB for 5 min. Hematoxylin was used for counterstaining. Immunostaining was observed using a ZeissAxio Imager A2 microscope and captured by IM500 software. Every experiment was repeated in three different biological replicates.

### 4.5. Statistical Analysis

All western blotting experiments were performed at least 4 times, and data were expressed as the mean ± SEM (for VEGF-A, VEGFR1, VEGFR2, ERK1/2, p38, cyclin D1, cyclin E1, and Cdc25A) and as the mean percentage (%) ± relative error (for p-VEGFR2/VEGFR2, p-ERK1/2/ERK1/2, and p-p38/p38 ratios). Comparisons were performed on the basis of age (LM vs. PM, and LV vs. PV) and parity status (LM vs. LV, and PM vs. PV). Experimental results of the molecular analysis were analyzed using ANOVA followed by the Bonferroni post-test. Results were considered statistically significant when *p* < 0.05. All statistical analyses were performed using the statistical package SigmaPlot v.11.0 (Systat Software Inc., San Jose, CA, USA).

## Figures and Tables

**Figure 1 ijms-24-03318-f001:**
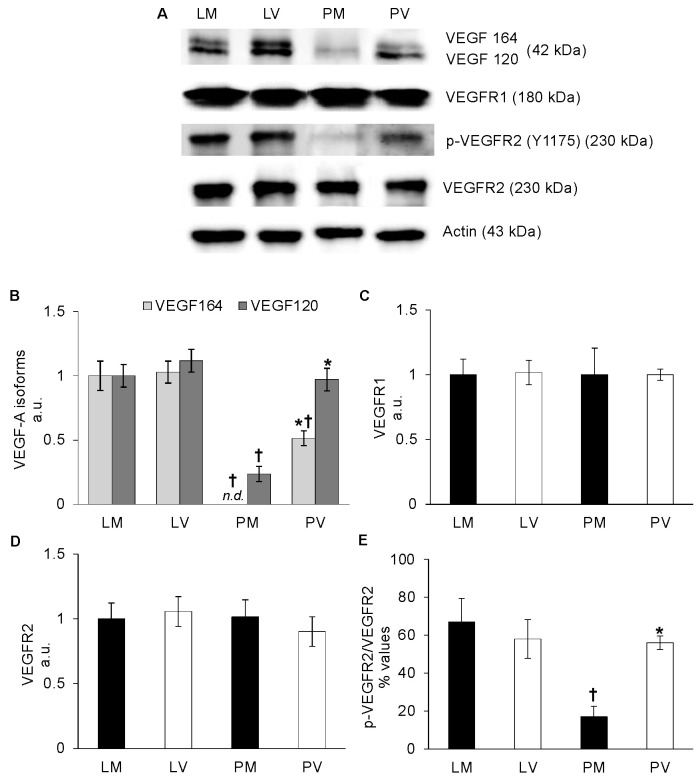
**VEGF-A and VEGFRs protein content in whole mice ovaries.** Representative western blot images of VEGF-A isoforms (VEGF 164 and VEGF 120), VEGFR1, and VEGFR2 total and phosphorylated form (p-VEGFR2,) of late-reproductive multiparous (LM) and virgin (LV), and post-reproductive multiparous (PM) and virgin (PV) mice (**A**). VEGF-A isoforms (**B**), VEGFR1 (**C**), and VEGFR2 (**D**) values are expressed as arbitrary units (a.u.), considering LM values arbitrarily as 1. The p-VEGFR2/VEGFR2 (**E**) values are expressed as percentages (%) of the ratio of phosphorylated/total protein. Bar graph data represent the mean ± SEM (**B**–**D**) and the mean percentage ± relative error (**E**) after normalization of each protein with the respective actin used as loading control of at least four independent determinations. (†) indicates significant difference (*p* < 0.05) related to age (LM vs. PM; LV vs. PV); (*) indicates significant difference (*p* < 0.05) related to parity status (LM vs. LV; PM vs. PV).

**Figure 2 ijms-24-03318-f002:**
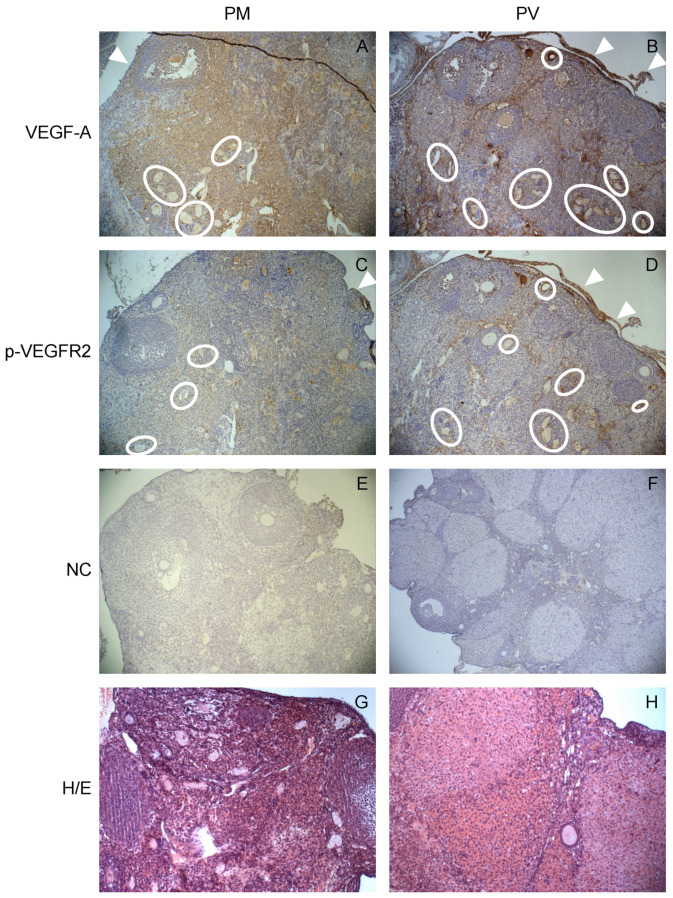
**Tissue localization of VEGF-A and p-VEGFR2 in mouse ovaries.** Representative images of VEGF-A (**A**,**B**) and p-VEGFR2 (**C**,**D**) immunoreactivity in post-reproductive multiparous (PM) and virgin (PV) ovaries. Negative controls (NC; **E**,**F**) and H&E staining (**G**,**H**) are presented. White circles indicate the presence of areas with small vessels; white arrowheads indicate the presence of ovarian surface epithelium, which appear to be multi-layered in PV ovaries (**B**,**D**). Magnification: ×100.

**Figure 3 ijms-24-03318-f003:**
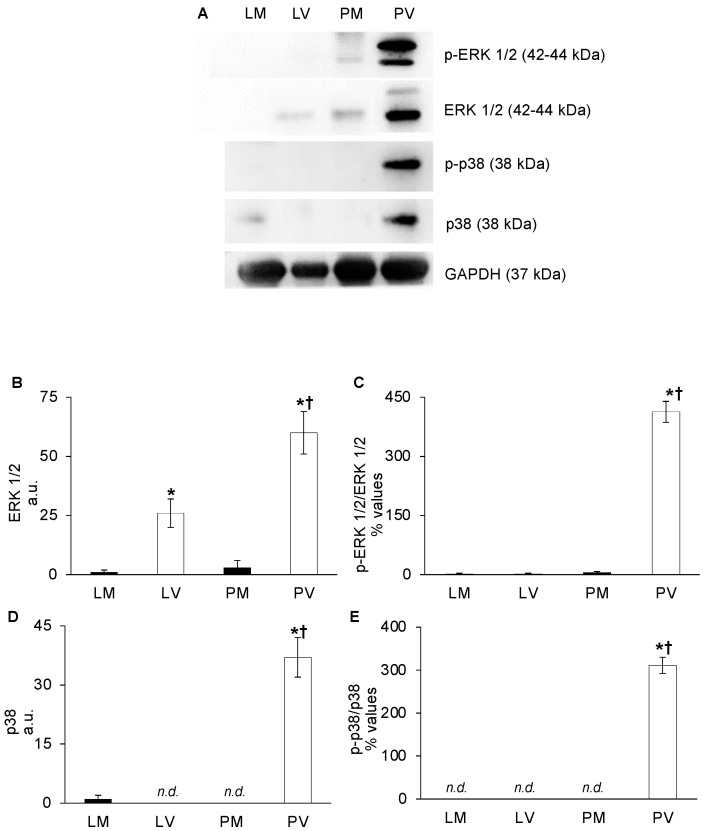
**VEGFR2 signaling pathway activation: total and phosphorylated ERK1/2 and p38 in whole mice ovaries.** Representative western blot images of ERK1/2 and p38 total and phosphorylated form (p-ERK1/2 and p-p38, respectively) of late-reproductive multiparous (LM) and virgin (LV), and post-reproductive multiparous (PM) and virgin (PV) mice (**A**). ERK1/2 (**B**) and p38 (**D**), values are expressed as arbitrary units (a.u.), considering LM values arbitrarily as 1. The p-ERK1/2/ERK1/2 (**C**), and p-p38/p38 (**E**) values are expressed as percentages (%) of the ratio of phosphorylated/total protein. Bar graph data represent the mean ± SEM (**B**,**D**) and the mean percentage ± relative error (**C**,**E**) after normalization of each protein with the respective actin used as loading control of at least four independent determinations. (†) indicates significant difference (*p* < 0.05) related to age (LM vs. PM; LV vs. PV); (*) indicates significant difference (*p* < 0.05) related to parity status (LM vs. LV; PM vs. PV).

**Figure 4 ijms-24-03318-f004:**
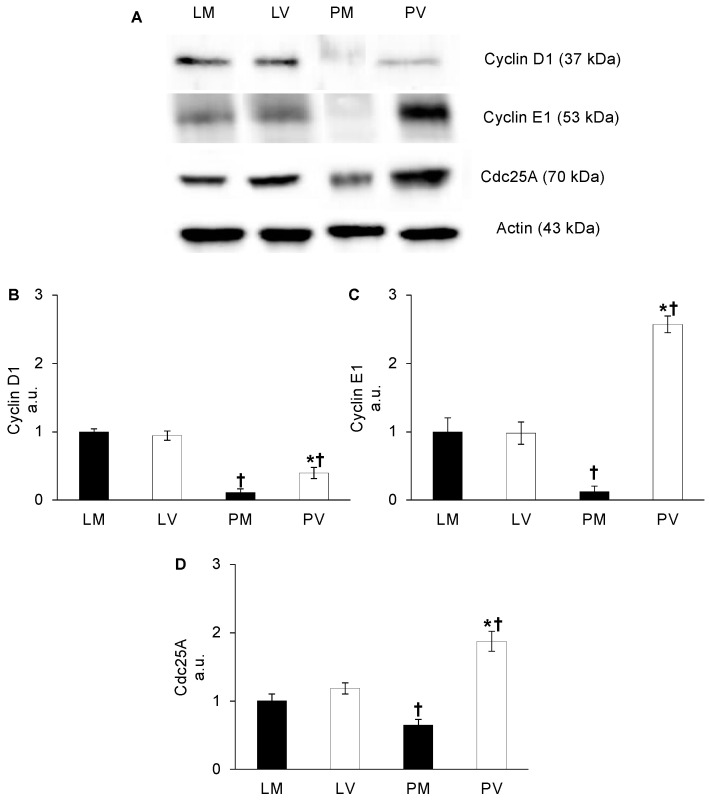
**Cell cycle regulator (cyclinsD1 and E1, and Cdc25A) protein content in whole mice ovaries.** Representative western blot images of cyclin D1, cyclin E1, and Cdc25A in late-reproductive multiparous (LM) and virgin (LV), and post-reproductive multiparous (PM) and virgin (PV) mice (**A**). Cyclin D1 (**B**), cyclin E1 (**C**), and Cdc25A (**D**) values are expressed as arbitrary units (a.u.), considering LM values arbitrarily as 1. Bar graph data represent the mean ± SEM after normalization of each protein with the respective actin used as loading control of at least four independent determinations. (†) indicates significant difference (*p* < 0.05) related to age (LM vs. PM; LV vs. PV); (*) indicates significant difference (*p* < 0.05) related to parity status (LM vs. LV; PM vs. PV).

## Data Availability

The data presented in this study are available within the article and upon request to the corresponding author.

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
