# Peer review of "Do Aging and Parity Affect VEGF-A/VEGFR Content and Signaling in the Ovary?—A Mouse Model Study"

_ijms, 2023, doi:10.3390/ijms24043318_

Round 1
Reviewer 1 Report
The manuscript entitled "VEGF-A/VEGFR2 expression is differentially regulated in late-reproductive and post-reproductive mouse ovaries in an age- and parity-dependent manner” is a novel, well-written, and valuable research paper.
First of all, I would like to emphasize that this study raises a very important issue regarding the health of modern women. As stated by the authors, research presented in this manuscript refers to the current widespread phenomenon in which women are delaying pregnancy into their 30s and beyond due to either social or personal reasons or because of infertility. This, unfortunately, can lead to an increased risk of gynecological cancers, particularly endometrial, breast, and ovarian cancers. In turn, the presented manuscript, as well as other research papers of the authors, allow us to conclude that the research of this team is one of many steps on the way to the physiological explanation of the aforementioned phenomenon.
I rate the reviewed paper very highly because the authors found an important and interesting research problem, extracted specific knowledge gaps, appropriately designed the experimental setup, clearly formulated the hypothesis and the aim of the study, and then skillfully matched the research methods to verify all the assumptions. The work is prepared very well: it is coherent, thoughtful, and provides clear "take-home messages". Nor do I doubt the scientific soundness of the authors; I believe that methodically, the entire experiment was carried out with great accuracy, dedication, as well as great substantive knowledge; I also appreciate a good knowledge of statistical analysis.
I have one major note to the entire experimental model (IMPORTANT REMARK below) and a few editorial notes or notes that require the rewording of selected sentences or the addition of specific information. In several places I suggest the authors analyze/rethink certain issues, or I give my suggestions for text editing, but of course, I do not require the authors to make these changes one-on-one; these are my attempts to hint to the authors (an attempt to facilitate/help).
I am very glad that I had the pleasure to read this manuscript, and also to have the opportunity to enrich the already high-quality paper with my small comments. I also hope that the research of this team will be further developed.
IMPORTANT REMARK: the ovary is a very diverse organ; stating that "samples" were collected is not sufficient, please specify in detail what were the samples, whether they were only selected structures, e.g. corpora lutea or were the whole ovaries were homogenized? This information should resonate well in the MATERIALS AND METHODS section.
EXPLANATION: in the further part of the review, the “->” sign will mean that I indicate editing the sentence from the form on the left side of the sign to the version on the right side of the sign.
TITLE
For a reader “being out of the loop” the title of this manuscript can be quite difficult to understand. In turn, the first sentence of the discussion highlights the aim of this work and I think that this thought, properly transformed, can be the best form of encouragement for the reader, and it will also highlight the strengths of the paper. Therefore, my title proposal is: “Do aging and parity affect VEGF-A/VEGFR expression and signaling in the ovary? - mouse model study” – this remark should be considered by the authors of the publication.
ABSTRACT
The abstract is to encourage further reading of the publication, but not to "attack" the reader with too much detail. So I would make two small changes:
LINE 14-15 – I think it is worth emphasizing the general purpose of the whole work here, i.e
“In this (…) were analyzed.” -> “In this study, the effects of aging and parity on VEGF-A/VEGFR expression and signaling in the mice ovaries were determined."
LINE 15-16 – “Nulliparous (virgins, V) … (post-reproductive, P).” -> “The research group consisted of nulliparous (virgins, V) and multiparous (M) mice during late-reproductive (L, 15 at 9-12 months and post-reproductive (P, 15-18 months) stages.”
INTRODUCTION
The authors very consistently refer to the process of angiogenesis throughout the manuscript, but this process is very extensive and is not based only on VEGF-A, VEGFR1, and VEGFR2; so it may be worth emphasizing why these specific angiogenic factors were selected, i.e. information such as:
VEGF-A is considered the key factor that stimulates endothelial cell proliferation, promotes cell migration, and induces stabilization of blood vessels (i.e. fundamental stages of vascular development); VEGF-A specifically binds to VEGFR1 and VEGFR2 receptors and thus triggers downstream ERK1/2 and AKT signaling; hence research on angiogenesis disturbances has been mostly focused on the VEGF-VEGFR signaling (as help: Angiogenesis and ovarian cancer, DOI 10.1007/s12094-009-0406-y; A peptide mimicking the binding sites of VEGF-A and VEGF-B inhibits VEGFR-1/-2 driven angiogenesis, tumor growth and metastasis, DOI:10.1038/s41598-018-36394-0).
Lines 49-50 – „some of the main downstream signaling molecules as ERK1/2, p38, cyclin D1, cyclin E1 and Cdc25A” -> this sentence sounds like "a random selection of factors", although after reading the whole manuscript I know that the proteins are selected for a specific purpose, but this information did not emerge at the beginning; I would divide these proteins into two groups that clearly emerge from the presented experiment, i.e. proteins related to VEGFR2 signaling pathway (ERK1/2, p38) and cancer targeting cyclin-dependent kinases (cyclin D1, cyclin E1, and Cdc25A), additionally two sentences from the discussion (lines 174-177 and 184-186; references 38.39, 40.41 and 45.46, respectively) relate to this information and could be included in this section.
In addition, I would specify the physiological role of the selected proteins, which I explain in more in the comments to the RESULTS section.
RESULTS
LINE 54 – in my opinion, the used methods do not allow us to conclude that "distribution and content" have been determined, and I would rephrase these phrases with "tissue localization and protein content"; and here a request to the authors to apply the change "distribution -> tissue localization" throughout the manuscript (also in the descriptions of figures), because in current literature data, phrase "distribution" refers to those analyzes where the target molecule is labeled and then it is checked where the protein is produced and where the protein is secreted and/or transported.
LINE 55 – similarly, a change of nomenclature regarding the experimental setup; "protein content", not "expression level"; while in the case of the receptors, protein expression and local concentration may be similar or even identical, in the case of secreted factors, such as VEGF-A, local concentration does not always coincide with local production; and here a request to the authors to change "expression -> protein content" throughout the manuscript, also in the descriptions of figures.
LINE 55 – specific isoforms of VEGF-A appear, and there was no proper information about this in the INTRODUCTION section; please briefly explain in the INTRODUCTION section why the authors considered these isoforms important to study; I understand that VEGF-A isoforms (120, 164, and 188) have distinct functions in vascular development and therefore specific isoforms may be crucial for the authors, but there is no information about it and of course it is worth emphasizing; I am sure that the authors have such knowledge, but in case of doubt, my suggestion: np.: Differential expression of VEGF isoforms in mouse during development and in the adult, DOI: 10.1002/1097-0177(2000)9999:9999<::AID-DVDY1093>3.0.CO;2-D
LINE 57 – „Conversely” -> „In turn, …”.
LINE 57 – „decrease of” -> „decrease in”.
LINES 58-59 – „Such a decrease was particularly evident” -> „The most noticeable decline was observed … ”.
LINE 62-63 – „significantly reduced (about -50%)” -> in fact, the decrease is significant, and I do not know if it is not more clear to indicate the multiplicity, i.e „significantly reduced (nearly two-fold)” - this remark should be considered by the authors of the publication.
LINE 97 – I would specify the title of the subchapter, "VEGFR2 signaling" -> "VEGFR2 signaling pathway activation", and similarly the description of Figure F3.
LINE 98 – „Although expressed almost exclusively in V mice” is a misleading statement because the reader thinks: what expression are authors talking about, is it VEGFR2 or signaling proteins expression?; please rewrite the sentence, it can be even two shorter sentences.
LINES 101-102 – it is always better to be very careful in Western Blot analyses; sometimes a low signal or no signal can be the result of the used reagents (I noticed that the authors of the manuscript are also careful in their judgments, which can be seen in the discussion) so I suggest that the sentence: „Remarkably, p38 kinase was expressed only in PV mouse, where it was entirely phosphorylated (PV vs PM, LV: P<0.05; Figure 3A,D,E).” rephrase to (for example): „It should be noted that p38 kinase (entirely phosphorylated) was reported only in PV mice, indicating a lack of target protein in other studied groups or too low detection capabilities of the used method.”
Line 115 – „cyclin D1, cyclin E1 and Cdc25” -> a situation similar to the VEGF-A isoforms, please provide a brief explanation in the INTRODUCTION section why the authors chose these proteins - it is about emphasizing the importance of these proteins in the form of one sentence or insertion; I assume authors mean cancer-related cell-cycle (may help: A review on the role of cyclin dependent kinases in cancers, DOI: 10.1186/s12935-022-02747-z; Targeting Cell-cycle Machinery in Cancer, DOI: 10.1016/j.ccell.2021.03.010).
LINE 122-123 – „and especially that of cyclin E was drastically reduced” -> „and especially the protein content of cyclin E drastically declined.”
LINE 125 – I propose to modify the subsection title -> „Cyclin-dependent kinases (cyclinsD1 and E1, and Cdc25A) protein content in mice ovaries.”
DISCUSSION
LINES 134-135 – as I also mentioned in the INTRODUCTION section, the angiogenesis process was not strictly studied in this study, but the influence of aging and parity on VEGF-A/VEGFR expression and signaling in the mice ovary; I think that this type of comparison (instead indicated sentence) should be included in the first sentence of the DISCUSSION section.
LINES 137-138 – “Unfortunately, we cannot assess VEGFR1 phosphorylation status due to the lack of commercially available antibodies targeting it.” -> this sentence refers directly to the used method and/or obtained results, so it should be included in the RESULTS section or MATERIALS AND METHODS section.
LINE 137 – „reduction” -> be more precise, i.e. decrease in the protein content/inhibition of phosphorylation/inhibition of the signaling pathway?
LINE 141 – „aging parity” -> I think a word is missing here, “aging and parity” or „parity/aging”?
LINES 161-162 – “Since altered VEGFR2 (…) the expression of ERK1/2 and p38 kinases.” -> this sentence gives a quite misleading message, activation of this receptor does not lead to the proliferation of all cells, it should be specified that we are talking about ENDOTHELIAL CELLS.
LINE 189 – “alarming marker” -> “possible alarming marker”.
LINE 195 – “exert” -> “may exert”, performed studies only indicate possible mechanisms, not prove them, so authors should be careful in their judgments.
MATERIALS AND METHODS
LINE 227 – „free access to food and water” -> „ad libitum access to food and water”.
LINE 245 – „resuspended” -> „immersed”.
LINE 257 – „antibodies conjugated to HRP, goat anti-rabbit…” -> „antibodies conjugated to HRP, i.e. goat anti-rabbit…” and a comma after “(1:5000)”.
REFERENCES
It seems that there are some errors in references (e.g. LINE 162, reference no 31), so please verify the bibliography in the entire manuscript.
Reviewer 2 Report
There seems to have problems in the WB for fig. 1 PM- VEGFR1and p-VEGFR2 lanes and in fig. 4 PM-cyclin D1, Cyclin E1.
In the original blots presented, there was no Molecular weight
markers to see whatever the size is and not a single loading control
is presented. Thus it is difficult to judge the quality and authenticity of the manuscript.
Round 2
Reviewer 2 Report
This reviewer has no concern. Can be published.